# Synergistic Cellular Responses Conferred by Concurrent Optical and Magnetic Stimulation Are Attenuated by Simultaneous Exposure to Streptomycin: An Antibiotic Dilemma

**DOI:** 10.3390/bioengineering11070637

**Published:** 2024-06-21

**Authors:** Jan Nikolas Iversen, Jürg Fröhlich, Yee Kit Tai, Alfredo Franco-Obregón

**Affiliations:** 1Department of Surgery, Yong Loo Lin School of Medicine, National University of Singapore, Singapore 119228, Singapore; nikolas.iversen@u.nus.edu; 2Institute of Health Technology and Innovation (iHealthtech), National University of Singapore, Singapore 117599, Singapore; 3BICEPS Lab (Biolonic Currents Electromagnetic Pulsing Systems), National University of Singapore, Singapore 117599, Singapore; 4Fields at Work GmbH, Hegibachstrasse 41, 8032 Zurich, Switzerland; juerg.froehlich@fieldsatwork.ch; 5Piomic Medical AG, Reitergasse 6, 8004 Zürich, Switzerland; 6NUS Centre for Cancer Research, Yong Loo Lin School of Medicine, National University of Singapore, Singapore 117599, Singapore; 7Competence Center for Applied Biotechnology and Molecular Medicine, University of Zürich, 8057 Zürich, Switzerland; 8Department of Physiology, Yong Loo Lin School of Medicine, National University of Singapore, Singapore 117593, Singapore

**Keywords:** wound healing, proliferation, magnetoreception, photomodulation, aminoglycoside antibiotics, magnetic mitohormesis, mitochondria, reactive oxygen species, chronic wounds, hard-to-heal wounds

## Abstract

Concurrent optical and magnetic stimulation (COMS) combines extremely low-frequency electromagnetic and light exposure for enhanced wound healing. We investigated the potential mechanistic synergism between the magnetic and light components of COMS by comparing their individual and combined cellular responses. Lone magnetic field exposure produced greater enhancements in cell proliferation than light alone, yet the combined effects of magnetic fields and light were supra-additive of the individual responses. Reactive oxygen species were incrementally reduced by exposure to light, magnetics fields, and their combination, wherein statistical significance was only achieved by the combined COMS modality. By contrast, ATP production was most greatly enhanced by magnetic exposure in combination with light, indicating that mitochondrial respiratory efficiency was improved by the combination of magnetic fields plus light. Protein expression pertaining to cell proliferation was preferentially enhanced by the COMS modality, as were the protein levels of the TRPC1 cation channel that had been previously implicated as part of a calcium–mitochondrial signaling axis invoked by electromagnetic exposure and necessary for proliferation. These results indicate that light facilitates functional synergism with magnetic fields that ultimately impinge on mitochondria-dependent developmental responses. Aminoglycoside antibiotics (AGAs) have been previously shown to inhibit TRPC1-mediated magnetotransduction, whereas their influence over photomodulation has not been explored. Streptomycin applied during exposure to light, magnetic fields, or COMS reduced their respective proliferation enhancements, whereas streptomycin added after the exposure did not. Magnetic field exposure and the COMS modality were capable of partially overcoming the antagonism of proliferation produced by streptomycin treatment, whereas light alone was not. The antagonism of photon-electromagnetic effects by streptomycin implicates TRPC1-mediated calcium entry in both magnetotransduction and photomodulation. Avoiding the prophylactic use of AGAs during COMS therapy will be crucial for maintaining clinical efficacy and is a common concern in most other electromagnetic regenerative paradigms.

## 1. Introduction

Recent advancements in wound healing technologies have underscored the potential of concurrent optical and magnetic stimulation (COMS) as a promising therapeutic modality for hard-to-heal wounds [1]. COMS has been shown to improve tissue perfusion and promote immunomodulation, thereby contributing to the healing process [2]. In the context of chronic wounds, diminished tissue perfusion and suppressed cellular respiration are major factors contributing to sustained inflammation that impairs the healing process. The COMS modality was devised with the aim of better promoting wound healing by improving cell proliferation, migration, vasodilation, and angiogenesis.

Extracellular Ca^2+^ entry and the production of reactive oxygen species (ROS) are the two most common responses observed across a broad range of electromagnetic stimulation paradigms [3]. In particular, the transient receptor potential canonical type 1 (TRPC1) cation channel has been implicated in calcium responses to electromagnetic field stimulation in muscle [4], neurons [5,6], and diverse other tissues [7,8,9,10]. Potentially uniting these two electromagnetic response limbs is evidence of a TRPC1–mitochondrial axis [4,11] that has been implicated in myoblast proliferation enhancement following electromagnetic exposure. Of importance, aminoglycoside antibiotics, commonly used in tissue culture and regenerative medicine paradigms, have been shown to impede Ca^2+^ permeation through TRPC1, potentially precluding the ability of TRPC1 to transduce magnetic stimulation into regenerative cellular responses.

This study sought to validate the demonstrated regenerative efficacy of COMS in a published in vitro myogenic regenerative model [4]. This study further aimed to examine the effect of aminoglycoside antibiotics on COMS cellular responses. Lastly, we aimed to explore potential functional synergism between the distinct excitation modes employed by COMS for improved mechanistic understanding of its mode of action and clinical translation.

## 2. Materials and Methods

### 2.1. Cell Studies and Analysis

#### 2.1.1. Tissue Culture

Murine skeletal myoblasts (C2C12) were purchased from the American Type Culture Collection (ATCC; LGC Standards, Teddington, United Kingdom) and were strictly maintained as previously described [4]. Briefly, C2C12 myoblasts were cultured in Dulbecco’s Modified Eagle Medium (DMEM) (Thermo Fisher Scientific, Waltham, MA, USA) supplemented with 10% fetal bovine serum (FBS) (Biowest, Nuaille, France), sodium pyruvate, and L-glutamine (Thermo Fisher Scientific, Waltham, MA, USA) and maintained in a humidified environment of 5% CO_2_. The myoblasts were seeded at a density of 3000 cells/cm^2^ into tissue culture dishes/flasks and, unless explicitly stated as part of an experimental protocol, were maintained in the absence of antibiotics at all times. Myoblast cultures were routinely subcultured every two days to maintain their confluence level below 40%. After 24 h of growth, cells were subjected to experimental interventions consisting of sham, optical, magnetic, or combined stimulation modes of the COMS. Cell quantification was evaluated following enzymatic dissociation using TrypLE Express Enzyme (Thermo Fisher Scientific, Waltham, MA, USA), followed by the standard trypan blue exclusion for viable cell determination and enumeration using a standard hemocytometer.

#### 2.1.2. Streptomycin Treatments

Streptomycin (100 mg/L; Merck KGaA, Darmstadt, Germany) was gently added in stages to the cell culture media 15 min before or after exposure to the distinct exposure conditions. Cell analyses were conducted 24 h following experimental intervention.

#### 2.1.3. Western Blot Analyses

Protein extraction from whole cell lysates was performed using ice-cold RIPA lysis buffer consisting of 50 mM NaCl, 1 mM EDTA, 50 mM Tris-HCl, 1% Triton X-100, 0.05% SDS (Sigma Aldrich, St. Louis, MO, USA), EDTA-free 1X protease inhibitor cocktail (Nacalai Tesque Inc., Kyoto, Japan), 1X PhosSTOPTM phosphatase inhibitor (Merck KGaA, Darmstadt, Germany), and 0.1% sodium deoxycholate (Merck KGaA, Darmstadt, Germany). Cell lysates were harvested using a cell scraper followed by their incubation at 4 °C for 30 min before being centrifuged at a speed of 3000 rpm for 10 min at 4 °C. The protein concentration was measured using Pierce BCA Protein Assay Kit (Thermo Fisher Scientific, Waltham, MA, USA). A total of 25 µg of protein extract supernatants were prepared in 4X Laemmli buffer (Bio-Rad Laboratories, Hercules, CA, USA) containing β-mercaptoethanol (Bio-Rad Laboratories, Hercules, CA, USA). The resulting extracts were heated to 98 °C for 5 min and resolved on a denaturing and reducing SDS-PAGE. The proteins were then transferred to a PVDF membrane (Thermo Fisher Scientific, Waltham, MA, USA) before blocking in 5% BSA (Thermo Fisher Scientific, Waltham, MA, USA) for 1 h. The antibodies and dilution factors utilized in the present study are provided in Table 1.

#### 2.1.4. Reactive Oxygen Species Measurements

To ascertain changes in ROS levels, myoblasts were plated into black-walled clear bottom 96-well plates (Corning Inc., Corning, NY, USA at a density of 5000 cells per well with replicates of 8 wells per condition. 24 h post-seeding, the cells were rinsed twice with PBS and then incubated with 5 µM of CM-H_2_DCFDA (Thermo Fisher Scientific, Waltham, MA, USA) in warm phenol-free and FBS-free FluoroBrite DMEM (Thermo Fisher Scientific, Waltham, MA, USA) for 30 min. The CM-H_2_DCFDA-containing media was removed by washing twice with FluoroBrite DMEM before exposure of the individual 96-well plates to the indicated COMS modalities for 5 min in a standard tissue culture incubator (see Appendix A). ROS measurement was performed using a Cytation 5 microplate reader (BioTek Instruments, Winooski, VT, USA) at Ex/EM: 492/525 nm for 15 readings up to 17 min.

#### 2.1.5. ATP Measurements

C2C12 myoblasts were seeded into clear 96-well plates at a density of 5000 cells per well in replicates of 3 wells per condition. After 24 h of cell growth, the cell plates were subjected to either sham (non-treated), light, magnetic fields, or COMS interventions for 5 min. The cell samples were then processed using a Cayman Chemical ATP Detection Assay Kit–Luminescence (700410; Ann Arbor, MI, USA) in accordance with the manufacturer’s instructions. Lysates were diluted 30-fold in 1 X ATP Sample Buffer before dividing each biological replicate into 2 technical replicates and loading into a white opaque 96-well plate. The ATP concentration was determined by luminescence reading using Cytation 5 microplate reader (BioTek Instruments, Winooski, VT, USA).

#### 2.1.6. Statistical Analyses

Statistical analyses were performed using GraphPad Prism (Version 10.2.0 for Windows, GraphPad Software, Boston, MA, USA). One-way analysis of variance (ANOVA) was used to compare the values between two or more groups supported by multiple comparisons. This was followed by Bonferroni’s post hoc test. A two-tailed *t*-test was used for comparisons between the mean of two independent samples.

### 2.2. Device

Customized COMS One devices were provided by Piomic Medical AG (Zürich, Switzerland). The COMS One is a Class IIa medical device, CE-certified for the stimulation of chronic leg and foot ulcers, which incorporates the technology for concurrent optical and magnetic stimulation.

Optical Stimulation Component: The COMS One features an optical stimulation system equipped with two sets of light-emitting diodes (LEDs) that emit at wavelengths of 660 nm (far-red) and 830 nm (near-infrared). The optical signals were pulsed at a frequency of 1 kHz with a maximum pulse width of 0.3 ms, delivering a peak power density of 25 mW/cm^2^ and an average power density of 5 mW/cm^2^ to the area of treatment. The COMS device in operation is shown in Appendix A.

Magnetic Stimulation Component: A coil system within the device produces pulsed modulated electromagnetic fields within the extremely low-frequency range of the electromagnetic spectrum. The emitted waveform is asymmetric and trapezoidal and is repeated at a frequency of 20 Hz. The peak field strength incrementally increases to 1.6 mT (16 Gauss), and treatments are administered over a 16 min session at the specified treatment location.

Experimental Setup: Three variations in the COMS One device were employed during the experiments:An optically functional model in which the magnetic stimulation component was deactivated;A magnetically functional model with the optical stimulation turned off;A fully functional unit operating as per the manufacturer’s specifications.

These configurations allowed for the assessment of the individual and combined effects of the optical and magnetic components.

## 3. Results

The proliferative effects of the separate and combined COMS modalities were examined in an established magnetoreceptive in vitro myogenesis system [4]. Exposure to light (red) and magnetic fields (blue) separately enhanced cell proliferation by 7% and 15%, respectively, relative to sham (black) (Figure 1). Exposure to combined light and magnetic fields (COMS modality; hatched blue/red) produced an increase in cell proliferation of 29%, which represented significant increases over light and magnetic fields individually. These results suggest that light and magnetic fields act synergistically to promote cell proliferation.

The administration of streptomycin, timed to coincide with magnetic exposure, was previously shown to preclude the transduction of magnetic fields into biological responses (magnetotransduction) [4]. To the best of our knowledge, the effect of streptomycin over photomodulation has not been examined prior to this study. Streptomycin added to the bathing media 15 min prior to exposure to any of the interventions attenuated cell proliferation (Figure 2B; Strep During Exposure). By contrast, the addition of streptomycin to the bathing media 15 min after the exposure to either light, magnetic fields, or the combination (Figure 2B; Strep After Exposure) was ineffective at inhibiting proliferation and, moreover, was indistinguishable from the responses of cells untreated with streptomycin (Figure 1). Therefore, a criterion for streptomycin antagonism of electromagnetic regenerative responses is that it be present at the time of exposure.

Cyclin D1 promotes cell cycle progression, favoring proliferation at the expense of differentiation [12]. Cyclin D1 protein levels were significantly increased over sham exposure (gray) by the combination of light and magnetic fields (COMS; hatched blue/red), whereas the other stimulation conditions did not significantly increase cyclin D1 protein levels (Figure 3A(i)). TRPC1 protein expression was also increased by the COMS modality (Figure 3A(ii)). This result agrees with previous reports of a TRPC1–mitochondrial signaling axis that is both responsible for, as well as a part of, the magnetoreceptive proliferative response invoking cyclin D1 [4]. In this respect, TRPC1 expression was previously shown to be upregulated by brief magnetic exposure, enhancing proliferation. Accordingly, TRPC1 expression and proliferation were conversely downregulated by the presence of streptomycin during magnetic exposure, which coincide with the previously reported feedback regulation of TRPC1 expression by magnetic field exposure [4]. The MAP/ERK pathway stimulates transcriptional cascades involved in cell proliferation and survival in response to magnetic field exposure [13]. Accordingly, Phospho-ERK protein expression was preferentially upregulated by COMS stimulation (Figure 3A(iii)). In agreement with cellular proliferative responses being antagonized by aminoglycoside antibiotics, streptomycin added 15 min before (and therefore, during) the comprehensive COMS stimulation precluded the upregulations of cyclin D1, TRPC1, and phospho-ERK, whereas the administration of streptomycin 15 min after COMS intervention did not (Figure 3B(i–iii)).

Changes in ROS and ATP production in response to the various interventions were next investigated. Paralleling the response in protein expression, cells treated with the COMS modality showed a significant reduction in ROS production (Figure 4A, absolute fluorescent intensity) relative to sham (black), whereas the reductions in ROS levels did not achieve statistical significance in response to the other exposure conditions, albeit showing the same incremental responses to light and magnetic fields as for cell proliferation (Figure 1 and Figure 2) and protein expression (Figure 3). Additionally, the rate of ROS accumulation showed a tendency to be reduced across all interventions relative to sham (Figure 4B). In an anti-parallel manner to ROS production, ATP levels incrementally rose from light to magnetic field exposures and achieved the greatest levels by the COMS intervention (Figure 4C). These results corroborate our other demonstrated indications of photon-electromagnetic synergism (Figure 1, Figure 2 and Figure 3) at the level of mitochondrial responses (Figure 4).

## 4. Discussion

Electromagnetism (pulsed magnetic fields and photons) is increasingly being recognized for its broad regenerative capacity if appropriately executed [3]. The separate and combined light and magnetic field components of the concurrent optical and magnetic stimulation (COMS) paradigm demonstrated graded and synergistic effects that served to modulate cellular respiration and that ultimately translated into enhanced proliferation. Light exposure on its own consistently produced the smallest change in a measured parameter yet acted in a supra-additive manner when combined with magnetic field exposure in the form of the COMS modality to produce the greatest overall response. Light, hence, appeared to play a faciliatory role in achieving synergism within the context of the comprehensive COMS stimulation.

In particular, TRPC1 has been implicated in both photomodulatory [14,15,16,17,18] as well as magnetoreceptive [4,5,11] biological responses. Members of the TRPC subfamily are the closest homologs to the *Drosophila* phototransductive TRP channel [19], the founder of the entire TRP superfamily [20,21,22]. Moreover, both magnetoreception [4] and photomechanical transduction [23] could be prevented by pretreatment with the same promiscuous TRPC1 antagonist, 2-APB. Aminoglycoside antibiotics also have been previously shown to disrupt TRPC1-dependent magnetoreception in diverse cell types [4,6]. Here, we examined the ability of streptomycin, an aminoglycoside antibiotic, to influence the photosensitive and magnetoreceptive components of COMS.

The addition of streptomycin to the bathing media 15 min prior to COMS stimulation attenuated a cellular response, whereas the addition of streptomycin 15 min following COMS stimulation did not interfere with cellular response to electromagnetic exposure (Figure 2). The critical difference was that streptomycin was present during cell exposure in the first scenario and did not coincide with exposure in the second scenario. Importantly, as the cell response was measured 24 h after electromagnetic exposure, the difference in time spent in the presence of streptomycin (30 min) was nominal by comparison. Hence, the observed differences in proliferation induction could not be attributed to the inhibition of protein synthesis commonly attributed to diverse antibiotic classes [24].

Cyclins D1 and B1 regulate cell cycle progression from G1 to S and from S to G2/M phases, respectively [25,26]. Both cyclins D1 and B1 are regulated by TRPC1-mediated calcium entry to control the proliferation of the same murine myogenic cell line employed in the present study [27,28]. Cyclins D1 and B1 expression levels were increased by all stimulation modes but most strongly by the COMS intervention. As previously reported, streptomycin added during exposure suppressed the COMS-induced rise in either cyclin, implicating the involvement of TRPC1 [4].

TRPC1 and TRPM7 are predominantly expressed in C2C12 myoblasts [4,27,29] as well as elsewhere in the body [20,30]. In support of TRPC1 serving to transduce electromagnetic stimuli, the expression pattern of TRPC1 coincided with log phase myoblast proliferation as well as sensitivity to magnetic field stimulation [29] and the silencing of TRPC1 expression, but not of TRPM7 expression, precluded proliferative responsiveness of myoblasts to magnetic exposure. Most critically, the vesicular reintroduction of TRPC1 into TRPC1 CRISPR-Cas9 knockdown myoblasts was necessary and sufficient to reinstate the magnetic induction of myoblast proliferation and mitochondrial responses [11]. Aminoglycoside antagonism of COMS cellular responses is likely occurring through the blockade of TRPC1-mediated calcium entry [31] that, in turn, serves to modulate magnetic mitohormetic (mitochondrial) adaptations [3,4,32,33]. The unified COMS modality most effectively reduced ROS and increased ATP levels. Such dichotomous changes in ROS (decrease) and ATP (increase) would be consistent with an improvement in mitochondrial respiratory efficiency [34] as a result of the synergistic actions of coincident light and magnetic field stimulations.

### 4.1. Aminoglycoside Antagonism of Other Calcium Channels

Calcium-permeable TRP channels are the primary sensory receptors of individual cells [20,35], whereas voltage-gated calcium channels relay stimuli reception between cells via direct electrogenic communication [3]. TRPC1 channels are hence predominantly expressed in proliferating progenitor cells [4,36,37,38,39], whereas voltage-gated calcium channels are more characteristic of differentiated tissues [4,36,37,38]. The aminoglycoside antibiotics also impede Ca^2+^ conduction via voltage-gated L-type calcium channels in diverse species and cell types [40,41,42]. However, the affinity of aminoglycoside antibiotics for the pore region of L-type voltage-gated calcium channels is much lower [41,42,43,44] than that for the pore region of TRPC channels [45], binding in the mM range rather than the µM range, respectively [46]. At the concentration of streptomycin administered in the present study (100 µg/mL) and based on developmental expression patterns of a variety of cation channels at the proliferative stage of myogenesis [4], TRPC1 is thus the most likely target for aminoglycoside antibiotic antagonism. On the other hand, the affinity series for the distinct aminoglycoside antibiotic species is similar for L-type voltage-gated calcium [41,42,43,44] and TRPC [45] channels.

### 4.2. COMS Electromagnetic Transduction

Evidence exists for both magnetic fields [4,5,11] and light [14,15,16,17,18] modulating TRPC1 function. Moreover, both blue (420 nm) [17] and red (635 nm) [18] diode laser stimulation are capable of invoking TRPC1-associated photomodulatory responses. Here, we showed that both magnetic field and light responses could be abrogated by streptomycin when applied coincidently with exposure, alluding to their mutual dependency on TRPC1. Moreover, the combination of light and magnetic fields (COMS modality) was capable of upregulating TRPC1 expression (Figure 2A) and could be attenuated with streptomycin (Figure 2B). Light given in isolation generally produced the smallest responses across measures, whereas when combined with magnetic field exposure, it produced significantly greater responses than magnetic field exposure alone, suggesting that light is permissive to magnetic sensitivity. This type of response would be consistent with the radical pair mechanism of magnetoreception [47], which has been shown to modulate mitochondrial energetics [48]. On the other hand, magnetic field-independent mechanisms for photomodulation have also been proposed that employ mitochondrial cytochrome c stimulation [49,50], possibly in combination with TRP-channel-mediated calcium entry [49,50]. At what level magnetic fields and photons impinge on TRPC1, as well as the relative contributions and form(s) of photomodulation that are invoked in the current exposure regimen, remain to be identified.

The objective of this study was to validate experimentally derived mechanistic features previously presumed to be involved in magnetoreception [3] in a clinically employed system. Key aspects of the electromagnetic signal such as magnetic field amplitude and symmetry, signal frequency and pattern, field directionality and uniformity, duration and frequency of exposure, wavelength, and source of photons will all influence the biological response and have been previously discussed in detail [3]. In the present study, no attempt was made to change discrete signal parameters in order to ascertain their individual biological contributions; instead, it focused on the potential AGA antagonism of electromagnetic signal transduction.

### 4.3. Clinical Implications of Aminoglycoside Antibiotic Antagonism of Electromagnetic Therapies

Aminoglycosides were amongst the first broad-spectrum antibiotics used for clinical purposes and are now garnering renewed interest due to the growing global incidence of antimicrobial resistance [51]. The antagonism of the COMS modality by aminoglycosides has importance beyond that of in vitro electromagnetic regenerative paradigms, as they are used across a broad array of in vitro and in vivo medical paradigms that are responsive to diverse developmental stimuli transduced by TRPC1 [3]. Their use will thus confound the interpretation of data generated by employing in vitro or in vivo electromagnetic paradigms. The topical use of aminoglycoside antibiotics is clinically appealing, as they can be administered locally at high concentrations for extended periods, effectively circumventing the risks associated with systemic usage, such as organ toxicity and gut microflora dysbiosis. Despite their broad acceptance, however, aminoglycoside antibiotics are minimally effective in treating deep-seated wound infections due to their poor penetration and uptake into persister cells [52,53,54]. Aminoglycosides are generally not the first line of defense in soft tissue infections, such as chronic or hard-to-heal wounds. Nonetheless, the prophylactic use of aminoglycosides in the treatment of other medical conditions, such as lung infection or tuberculosis, will undermine the regenerative efficacy of COMS-related methodologies.

## 5. Conclusions

The distinct electromagnetic stimulation modes of concurrent optical and magnetic stimulation (COMS) were shown to exert incremental yet synergistic effects. Light appeared to play a permissive role when combined with extremely low-frequency magnetic field exposure, resulting in supra-additive responses. COMS-induced changes in cell proliferation, cell cycle regulatory proteins, and mitochondrial respiration were correlated with TRPC1 channel expression, previously reported to participate in a calcium–mitochondrial axis invoked by magnetic fields and antagonized by aminoglycoside antibiotics. Accordingly, streptomycin hampered the ability of the separate and combined COMS stimulation modalities to induce proliferative responses. However, at the dose of streptomycin used in the present study (100 µg/mL), the synergism between concurrent light and magnetic stimulation was capable of partially overcoming the antagonism of streptomycin to achieve statistically significance proliferative enhancements. Given the broad developmental attributes of TRPC1 offset by its propensity to be antagonized by aminoglycoside antibiotics, the coincident use of aminoglycosides during regenerative medicine paradigms, including electromagnetic modalities, should be carefully considered.

## Figures and Tables

**Figure 1 bioengineering-11-00637-f001:**
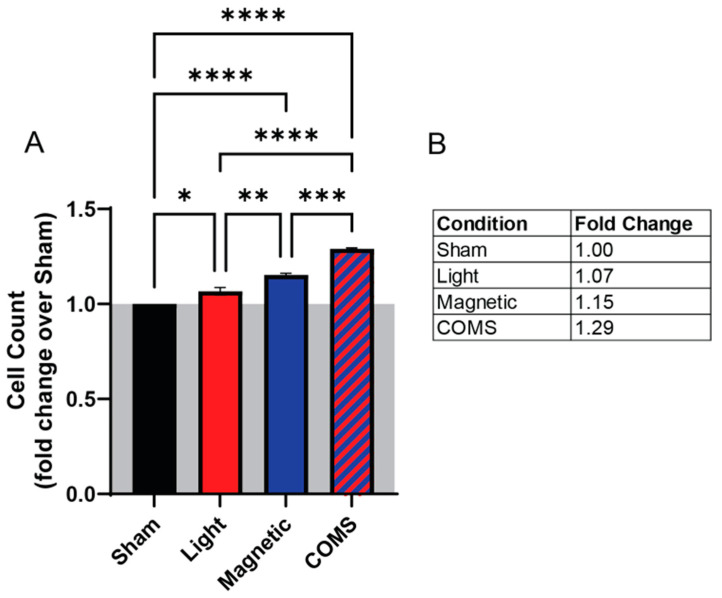
**Incremental myoblast proliferation induction upon exposure to light, magnetic fields, or their combination in the absence of streptomycin.** (**A**) Live cell count of murine C2C12 myoblasts in response to sham (black), light (red), magnetic fields (blue), or combined COMS (hatched blue/red) exposure. (**B**) Table of fold changes in live cell count relative to the sham condition. The gray shaded area represents the absence of streptomycin at all times during exposure to the indicated conditions. Statistical analyses were performed minimally in three independent biological replicates. Data were analyzed using one-way ANOVA followed by multiple comparison tests. Significance levels are indicated as follows: * *p* < 0.05, ** *p* < 0.01, *** *p* < 0.001, and **** *p* < 0.0001. Error bars represent the standard error of the mean (SEM).

**Figure 2 bioengineering-11-00637-f002:**
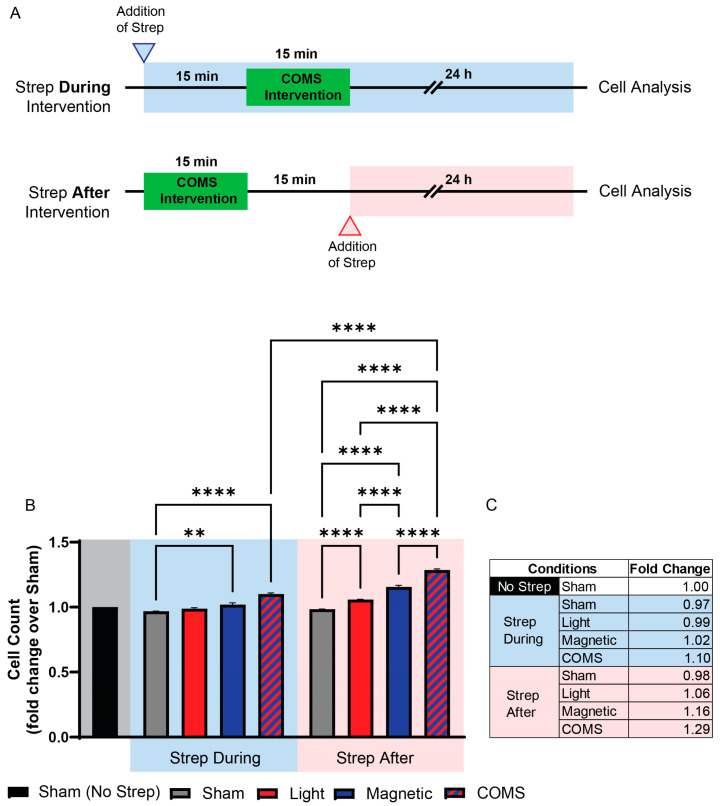
**The timing of streptomycin administration produced differential effects over myogenic proliferation induction in response to light and magnetic field combinations.** (**A**) Schematic illustration of when streptomycin (100 µg/mL) was added to the bathing media of cells in the “Strep During” (blue shaded) and “Strep After” (red shaded) paradigms prior to cell enumeration at 24 h after the indicated interventions. (**B**) Live cell count of mouse murine myoblasts in response to the different stimuli as aforementioned with streptomycin antibiotic supplementation (100 µg/mL) added 15 min before (left side, blue shaded box) or after (right side, red shaded box) exposure to the intervention. (**C**) Table showing the fold change of live cell count over sham condition (without streptomycin administration). The shaded areas indicate the absence of streptomycin (gray) or its application before and during (blue) or after exposure (red) to the indicated conditions. All data collected were from cells of the same plating. Statistical analyses were performed minimally in three independent biological replicates. Data were analyzed using one-way ANOVA followed by multiple comparison tests. Significance levels are indicated as follows: ** *p* < 0.01, and **** *p* < 0.0001. The error bars represent the standard error of the mean (SEM).

**Figure 3 bioengineering-11-00637-f003:**
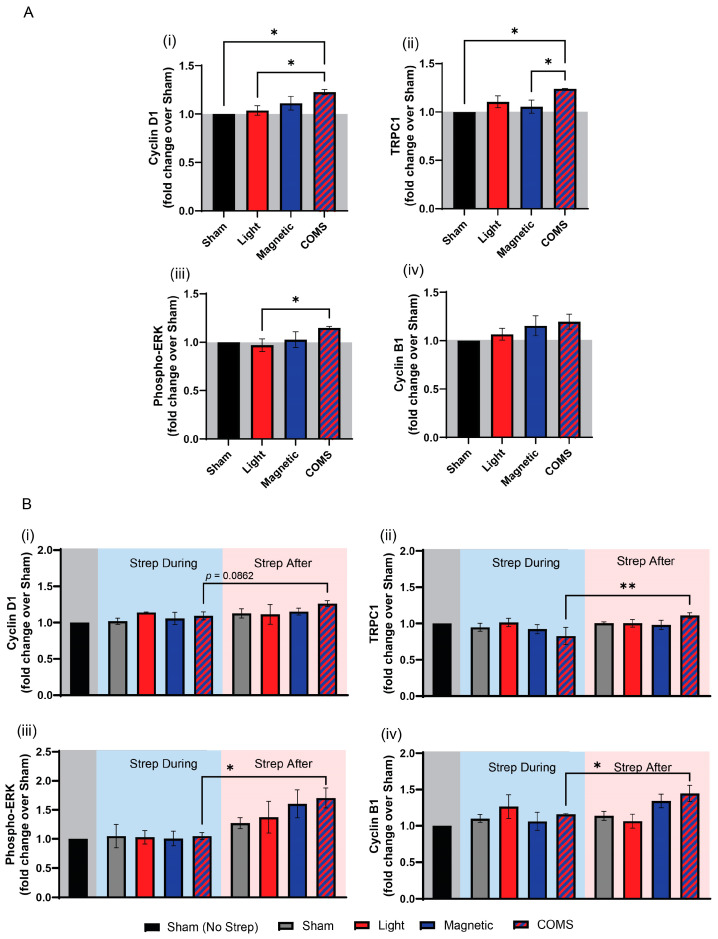
**Myogenic proliferation associated protein expression in response to exposure to light, magnetic fields, and their combination.** (**A**) Protein expression of (**i**) cyclin D1, (**ii**) TRPC1, (**iii**) phosphorylated ERK, (**iv**) and cyclin B1 in response to the indicated exposure intervention (n = 3) in the absence of streptomycin. (**B**) Protein expressions of (**i**) cyclin D1, (**ii**) TRPC1, (**iii**) phosphorylated ERK, (**iv**) and cyclin B1 either in the presence of streptomycin (100 µg/mL) during (blue shaded box) or after (red shaded box) exposure as indicated (n = 3). The shaded areas indicate the absence of streptomycin (gray) or its application before and during (blue) or after exposure (red) to the indicated conditions. All data shown in panel (**B**) were collected from cells of the same plating and represent independent cell samples as those used in panel (**A**). Statistical analyses were performed minimally in three independent biological replicates. Data were analyzed using one-way ANOVA followed by multiple comparison tests. Significance levels are indicated as follows: * *p* < 0.05 and ** *p* < 0.01. The error bars represent the standard error of the mean (SEM).

**Figure 4 bioengineering-11-00637-f004:**
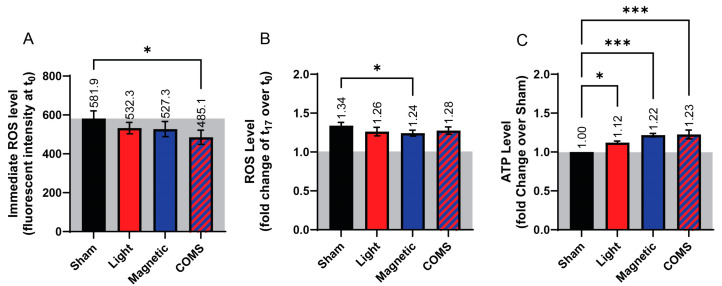
**Dichotomous changes in ROS and ATP production following COMS exposure**. (**A**) Bar chart showing the immediate ROS response of cells (fluorescent intensity averaged during initial 3 min of reading) for the indicated interventions (n = 4). (**B**) ROS level of cells expressed as fluorescent intensity fold change at 17 min (t_17_) over time 0, (t_0_), (n = 5). (**C**) The bar chart shows the ATP levels of cells (expressed as fold change over Sham) at t_17_ (n = 5, with six technical replicates each). In all cases, cells were exposed for 5 min to the indicated exposure modality at the start of device activation before measurements were commenced. The presented data were generated in the absence of streptomycin. Data were analyzed using one-way ANOVA followed by multiple comparison tests with * *p* < 0.05 and *** *p* < 0.001. The error bars represent the standard error of the mean (SEM).

**Table 1 bioengineering-11-00637-t001:** List of antibodies, vendors, and dilution factors.

Protein Target	Vendor, Country	Dilution Factor
Cyclin D1 (CD1)	Santa Cruz Biotechnology, Dallas, TX, USA	1:300
Transient receptor potential canonical 1 (TRPC1)	Santa Cruz Biotechnology, Dallas, TX, USA	1:500
Glyceraldehyde-3-phosphate dehydrogenase (GAPDH)	Proteintech Group, Inc., Rosemont, IL, USA	1:10,000
Phosphorylated ERK and Total ERK	Santa Cruz Biotechnology, Dallas, TX, USA	1:300

## Data Availability

All data supporting the results are presented in the manuscript. Any other inquiries can be directed to the corresponding authors via email.

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
