# Peer review of "Synergistic Cellular Responses Conferred by Concurrent Optical and Magnetic Stimulation Are Attenuated by Simultaneous Exposure to Streptomycin: An Antibiotic Dilemma"

_bioengineering, 2024, doi:10.3390/bioengineering11070637_

Round 1

Reviewer 1 Report

Comments and Suggestions for Authors

Synergistic Responses Conferred by Concurrent Optical and Magnetic Stimulation are attenuated by Simultaneous Exposure to Streptomycin: An Aminoglycoside Antibiotic Dilemma.  The authors expand on their previous research in the paradigm of magnetic field stimulation within the context of reactive oxygen species and calcium signaling.  Here, the addition of light stimulation with magnetic fields adds an extra dimension to improve cellular responses, called COMS.  The authors have previously shown a link between ROS and Ca2+ signaling involves the TRPC1 and aminoglycoside antibiotics attenuated magnetic field responses.  This current paper builds on the emerging paradigm (reference 4) that electromagnetic stimulation imparts a magnetic mitohormesis effect and should avoid the use of AGAs to maximize response in these types of experiments and therapies.

Overall, the set of experiments appears to show the combined effect of magnetic and light stimulation on a variety of cell assays.  However, the authors also attempt to integrate their previous hypothesis of AGAs as TRPC1 antagonists that is woven throughout the summary.  I think it would be beneficial for the authors to clarify the points below in the manuscript.

Line 218-234 Figure 3 caption and results section is confusing.  For example, it seems that 3A is no strep added and 3B is the administration time dependence and impact of strep i.e. before or after light, magnetic, and COM.  It would be helpful in both the text and figure caption to state the 3A is a strep independent experiment.  At this point, I am not sure if it is or isn’t.  For example, Line 224 sates that TRPC1 confirms previous results in ref 4, but the next sentence “TRPC1 expression also appeared to be most sensitive to streptomycin antagonism, which coincides with demonstrated feedback regulation of TRPC1 expression by magnetic field exposure” appears to imply this was part of the experiment (strep was included) and also confirming ref 4.  In my opinion, I think the authors are intending to highlight that TRPC1 was also part of strep antagonism but Fig 3A isn’t a strep experiment.     

After re-reading Figure 1 results, it would also be helpful to make clear that figure 1 experiments are strep free and the only strep experiments are the before and after timed experiments.  I think this is true based on the experimental procedure section on “strep treatments”.

Line 247-249 If these results corroborate previous indications of synergism… then a citation should be referenced.

Figure 4, I believe these are also strep free experiments but not clear.

Minor comments:

Line 88: CO2

Line 90: cells/cm2

Line 99:  “distinct paradigms” – perhaps “conditions” instead of paradigms.

Line 119: there should be a comma after ROS

Line 122-23: CM-H2DCFDA 

Comments on the Quality of English Language

n/a

Reviewer 2 Report

Comments and Suggestions for Authors

1-      Since similar research has already been published, the authors should highlight the work's novelty and how it differs from those prior studies.

2-      Please remember that mW/ cmrepresents power density rather than average power or peak power, thus please take this into account throughout the entire document.

3-      What effects should be expected if the optical signals are pulsed at a different repetition rate other than 1 kHz? Please discuss this issue in the text.

4-      What effects should be expected if the optical signals are at a different pulse width of 0.3 ms? Please discuss this issue in the text.

5-      What effects should be expected if the irradiation time was varied? Please discuss this issue in the text.

6-      What effects should be expected if the irradiation power density was varied other than 25 mW/cm2? Please discuss this issue in the text.

7-      What effects should be expected if the irradiation light wavelength was varied?

8-      What effects should be expected if laser light was used instead of light emitting diode?

Comments on the Quality of English Language

Moderate editing of English language required

Round 2

Reviewer 2 Report

Comments and Suggestions for Authors

The manuscript has been satisfactorily revised by the authors in response to my earlier remarks and concerns. Overall, the manuscript flows smoothly and makes the authors' work more understandable. In my opinion, the manuscript contains now all the information and is suitable for publication in Bioengineering.